# Effects of wood fiber impulse-cyclone drying process on the UV-accelerated aging properties of wood-plastic composites

**Qingde Li[1], Feng Chen[2]\*, Tonghui Sang[3]**

**1** School of Industrial Design and Ceramic Art, Foshan University, Foshan, Guangdong, China, **2** School of Art and Design, Taizhou University, Taizhou, Zhejiang, China, **3** Faculty of Creative Technology and Heritage, Universiti Malaysia Kelantan, Pengkalan Chepa, Kelantan, Malaysia

☉ These authors contributed equally to this work.
* chenfeng1984@tzc.edu.cn

**Data Availability Statement:** All relevant data are within the paper and its Supporting Information files.

## Abstract

The impulse-cyclone drying and the silane coupling agent (A187) modification are applied to treat wood fibers under the following conditions: 180˚C, 180˚C+A187, 200˚C+A187, 220˚C +A187 and 240˚C+ A187. Then, HDPE/wood fiber composites are fabricated with a two-stage plastic extruder, and the effects of impulse-cyclone drying technique on the UV-accelerated aging properties of composites are investigated. Fourier-transform infrared spectroscopy (FTIR) reveals that the silane coupling agent chemically reacts with the hydroxyl groups on the wood fiber surfaces, the anti-UV aging properties of composites is enhanced. Mechanical test shows that during the 0–3000 h of UV aging process, the mechanical properties of samples tend to increase initially and then decrease within a period of time. After 3000 h of UV aging, the specimen 4 exhibits the least loss of mechanical properties, with flexural modulus, flexural modulus and impact strength of 65.40 Mpa, 2082.08 Mpa and 12.85 Mpa, respectively. The effects of impulse-cyclone drying technique on the UV-accelerated aging properties of composites are investigated through Spectrophotometry and Surface microstructure observation. indicates that the ΔL* and $\Delta E$* values increase greatly at the stage of 0–1000 h aging, which though tend to stabilize after 1000 h. The degree of discoloration changes little for specimen 4,and the number of surface cracks is relatively small, which exhibits the optimal aging resistance. In conclusion, the addition of wood fibers treated by impulse-cyclone drying (220˚C) and A187 modification is effective in enhancing the anti-UV aging properties of HDPE/wood fiber composites. Nevertheless, such enhancing effect turns to decline when the temperature of impulse-cyclone drying treatment is excessively high.

## Introduction

In harsh natural environments, the composites from biomass are easily damaged, which undergo changes in durability and substrate structure [1, 2], as well as biodegradation [3, 4].

**Funding:** The National Natural Science Foundation of China (Grant No.31901243) financially supported this research.

**Competing interests:** The author(s) declared no potential conflflicts of interest with respect to the research, authorship, and/or publication of this article.

Through UV aging, these eventually lead to decreases in the mechanical properties, weather resistance and thermal performance of composites [5, 6]. Hence, improving the composite performance seems critical.

Regarding research on the mechanical properties and weather resistance of biomass composites, Nicole M. et al. used porous silicon foams to fabricate reinforced high-density polyethylene (HDPE) foamed composites, thereby substantially improving the mechanical properties, thermal properties and crystallinity of the HDPE foamed composites while imparting certain brittleness to them [7]. Feihong Liu et al. investigated the hygroscopic properties of south yellow pine/HDPE composites, compared the hygroscopic and desorptive properties of wood-plastic composite (WPC) specimens with Nelson adsorption isotherms under the same temperature and humidity conditions, and developed a relevant prediction model [8]. Substituting RPET for LCP, Supattra Kayaisang et al. fabricated thermoplastic composite fiber materials with excellent reinforcement and high heat resistance. However, their study did not further discuss the enhancement of mechanical properties [9]. Boonsri Kusuktham et al. reinforced WPCs using calcium silicate, which improved the mechanical properties and weather resistance, albeit certain irritation property [10]. Eleftheria Roumeli et al. fabricated a reinforced hemp/HDPE composite using maleic anhydride. Despite substantially improved material properties and largely enhanced mechanical and thermal properties of the composite, the cost was rather high [11]. Based on the response surface analysis, Ramachandran [12] and Anjana et al. [13] optimized the machining process of PP/HDPE/nano-kaolinite clay composite. Their results indicated that the optimization of process parameters improved the thermal stability and dynamic thermomechanical properties of the composite. However, they failed to probe deep into the weather resistance of composite material. Sanchez-Valdes et al. used a twin-screw extruder to melt process a montmorillonite mixture modified with polyethylene and ammonium-based surfactant. Their results showed that all the compatible nanocomposites were preferably viscous, and that higher clay contents would result in poor clay dispersion or intercalation effect, insignificant changes in melting temperature, as well as an evident reduction in crystallinity [14]. Sibeko et al. explored the effects of vinyltriethoxysilane treatment and nanoclay content on the low-density polyethylene (LDPE)/clay nanocomposite. They found fundamentally unchanged thermostability of the composite, and altered dynamic mechanical and tensile properties of LDPE by the presence of vinyltriethoxysilane and nanoclay [15]. M. A. Nour et al. modified clay with phenol formaldehyde silane resin, which was then injected into the HDPE composites (3, 5 and 7 wt%). The Fourier-transform infrared spectroscopy (FTIR) spectra indicated successful introduction of amino groups into the HDPE structure, while the cone calorimetry revealed reduction of heat release rate, prolonged composite ignition and enhanced aging resistance [16].

Focusing on the WPC fabricated with impulse-cyclone dryer, this study investigates the high-temperature hot air treatment conditions (initial moisture content, airflow temperature, airflow velocity and loading rate) on the modification efficiency of biomass fibers, and explores the effects of fiber modification conditions on the UV-accelerated aging, mechanical properties, chromaticity, dynamic thermomechanical (DMA) properties and thermogravimetric (TG) behavior of WPCs. Correlations between the high-temperature hot air modification conditions and the mechanical properties of WPCs are established, and relevant mechanisms are revealed.

## Materials and methods

### Materials

Wood fibers: Poplar residues were extracted via a wood fiber pulverizer (Model 60, Fuyang Energy Technology Co., Ltd., Xuzhou, Jiangsu, China). The poplar microfibers were then

**Table 1. Performance indicators of poplar fibers before and after drying treatment.**

| Performance indicator | Before drying | After drying |
|---|---|---|
| *Fiber morphology/mesh* | 60–80 | 60–80 |
| *Fiber length/mm* | 1.36(0.20) | 1.34(0.11) |
| *Fiber diameter/μm* | 223(31) | 211(21) |
| *Length–diameter ratio* | 6.1(1.87) | 6.3(0.67) |
| *Moisture content/%* | 10.6(0.19) | 2.3(0.09) |

Note: The results are means of measurements, and the numbers in brackets are the sample variances.

subjected to an impulse-cyclone dryer (Model MQG-50, Jianda Drying Equipment Co., Ltd., Changzhou, China) to prepare well-dispersed poplar fibers. Table 1 lists the performance indicators of poplar fibers prior to and after drying (S1 Table in S1 File). High-density polyethylene (HDPE), with a density of 954 kg·m⁻³, a melting index of 0.7 g/10 min, a crystallinity of 85% and a molar mass of $3\times10^5$, was purchased from CNPC Daqing Petrochemical Co., Ltd. Meanwhile, Silane coupling agent (A187) was procured from Benchmark Chemical Reagent Co., Ltd., Tianjin.

## Methods

**Impulse-cyclone drying treatment of fibers.** As shown in Fig 1, the impulse-cyclone drying treatment featured short duration, high strength, high efficiency, simple process and environmental friendliness (S1 Fig in S1 File). The thermal drying conditions were drying temperatures of 160°C, 180°C, 200°C, 220°C, 240°C, a dryer air velocity of 11 m/s, a feeding rate of 120 kg/h, and a fiber morphology of 60–80 meshes. The fibers in stable stage were

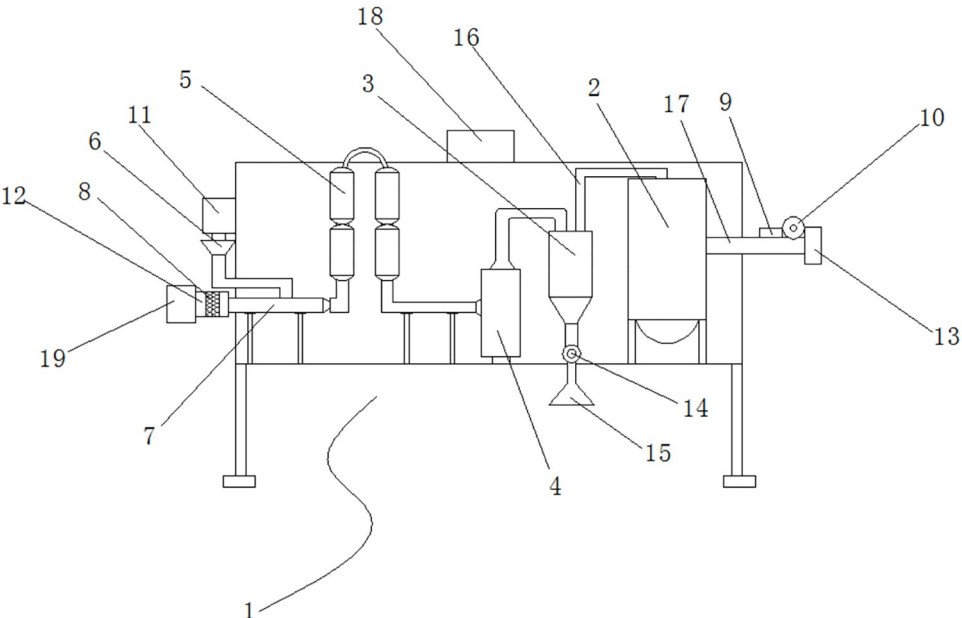

**Fig 1. Impulse-cyclone dryer.** 1. Intallation body; 2. Impulse bag filter; 3.Cyclone separator; 4. Cyclone dryer; 5. Impulse dryer; 6. Helical feeder; 7. Air inlet pipe; 8. Air filter; 9. Air regulating valve; 10. Fan; 11. Feed port; 12. Air inlet port; 13. Exhaust port; 14. Closed air discharger; 15. Discharge port; 16. Branch pipe; 17. Exhaust pipe; 18. PCL controller; 19. Heater.

removed for sampling, which had final moisture content of 1–3%, and the samples were sealed in vacuum bags.

**Silane coupling agent treatment of dry fibers.** Absolute ethanol solution was mixed with acetic acid at a 9:1 ratio while controlling the pH at 3–4, which was then added with silane coupling agent (A187) and magnetically stirred till reaching a target concentration of 5% and homogeneity. The stirring lasted for about 30 min. During mixing, a homogenizer was used for stirring operation and spraying method was implemented till the coupling agent and the wood fibers were mixed completely and evenly. The treated fibers were then dried in a 120˚C oven for 2 h, so that the ethanol and the acetic acid were almost completely evaporated.

**FTIR characterization.** A Fourier transform infrared spectrometer (Magna-IR560, Nicolet, Wisconsin, USA) was utilized to examine the group changes before and after modification of wood fibers. The test samples were mixed with KBr powder, and then pressed into thin discs for FTIR analysis over a wave number scanning range of 4000–500 cm$^{-1}$ at a 4 cm$^{-1}$ resolution and a 32 times/min scanning frequency [17].

**Preparation and testing of UV-accelerated aging specimens.** The impact of UV-accelerated aging on the HDPE/wood fiber composites in outdoor environment was simulated according to the QUV test procedure. The specimens (76.2 mm × 76.2 mm × 3 mm) were placed into an accelerated weathering tester (QUV/SPRAY, Q-Panel Lab Products, Ohio, USA), and processed under an aging program set in accordance with ASTMG-154. The specific parameters are detailed in Table 2.

Each cycle of aging process lasted for 12 h, which was divided into two stages. In the first stage [18], the damage caused by sunlight was simulated using UV fluorescent tubes. The composites were irradiated by UV light with a 340 nm wavelength at an intensity of 0.77 W·m$^{-2}$ while controlling the UV exposure temperature within the tester at 50˚C. The second stage [19] was a 4 h course of circulating condensation, during which the temperature in the tester was controlled at 40˚C for simulating the impact of outdoor humidity on the composites. During the circulating condensation, water was heated with a heating device located below the water tank, so that the testing chamber was filled with steam, where the humidity could reach 100% upon equilibrium. The vents on top of the water tank ensured that the condensed steam was filled with oxygen, allowing continuous condensation of steam on the testing plate and specimens, which then flew back to the water tank. Performance tests were carried out after 500 h, 1000 h, 1500 h, 2000 h, 2500 h and 3000 h of accelerated aging, respectively, and the test results were expressed as the means of five specimens.

**Preparation and testing of chromatic specimens.** A spectrophotometer (CM-2300d, Konica Minolta, Japan) was utilized to measure the surface chromaticity values of HDPE/wood fiber composite specimens. The $L^*a^*b^*$ color system developed by the International Commission of Illumination CIE (1976) was used for color notation, where the data comprised luminosity values ($L^*$) and two chromaticity coordinates ($a^*$ and $b^*$). The changes in color could be expressed by Formulas (1) and (2) [20]. Five specimens (80 mm×13 mm×4 mm) were tested per period over a wavelength range of 360–700 nm, each at three points, which totaled 15 times of tests, and the test results were expressed as means. The computational formulas for $\Delta E^*$ and $\Delta L^*$ were as follows [21, 22]:

$$\Delta E^* = (\Delta L^{*2} + \Delta a^{*2} + \Delta b^{*2})^{1/2} \tag{1}$$

**Table 2. UV-accelerated aging test conditions.**

| Stage | Duration (h) | Temperature(˚C) | Wavelength (nm) | Irradiation intensity (W·m$^{-2}$) |
|---|---|---|---|---|
| *UV aging stage* | 8 | 50 | 340 | 0.77 |
| *Condensation stage* | 4 | 40 | — | — |

$$\Delta L^* = L^* - L_0^* \tag{2}$$

$$\Delta a^* = a^* - a_0^* \tag{3}$$

$$\Delta b^* = b^* - b_0^* \tag{4}$$

where $\Delta E^*$——degree of discoloration;

$\Delta L^*$——degree of luminosity change;

$L^*$——luminosity index, with a variation range of 0–100, where the completely white objects were regarded as 100, and the completely black objects were regarded as 0. A larger value indicated that the luminosity of specimen increased and the surface became brighter. Conversely, a smaller value meant that the specimen darkened;

$a^*$——red–green chromaticity index, with a variation range from -150 to 150. An increase in the $a^*$ value indicated that the specimen color gradually shifted towards the red direction, while a decrease indicated gradual shift of color towards the green direction;

$b^*$——yellow–blue chromaticity index, with a variation range from -150 to 150. An increase in the $b^*$ value indicated that the specimen color gradually shifted towards the yellow direction, while a decrease indicated gradual shift of color towards the blue direction.

**Preparation and testing of mechanical properties specimens.**   A universal mechanical tester (WDW-20, Kexin Testing Instrument, Changchun, Jilin, China) was utilized to measure the mechanical properties of HDPE/wood fiber composites before and after accelerated aging. Through three-point bending test, the flexural strength of specimens (80 mm×13 mm×4 mm) was measured as per ASTM D 790 [23] across a span of 80 mm (16 times the thickness of specimens) at a loading speed of 2.5 mm/min. Impact strength was measured through the simply supported beam pendulum impact test in accordance with ASTM D256 [24]. The specimen dimensions were 80 mm×10 mm×4 mm, while the pendulum impact tester (XJC-25, Chengde Precision Testing Machine Co., Ltd.) was set to a span of 60 mm, a pendulum energy of 2 J, and an impact velocity of 2.9 m/s. The test results were expressed as the arithmetic means of five specimens. A scanning electron microscope (QUANTA200, FEI, Netherlands) was utilized for the material surface characterization and the test results analysis.

**Microstructure analysis.**   The working principle of scanning electron microscope (SEM) was based on the interaction between electrons and matter. To be specific, when a beam of high-energy incident electrons bombarded a material surface, it acquired various information about the physical and chemical properties of sample itself by exploiting the electro–matter interaction. During the experiment, the samples were prepared into smaller blocks with cutter, placed onto the sample holder while trying to keep the upward surfaces smooth. After firmly fixing both sides of samples with carbon conductive adhesive, the samples were sputtered with gold layer in a vacuum coater. The experimental process was implemented in accordance with ASTM E1588 (2017) [25], and the sample morphological characteristics were observed using QUANTA 200 SEM system under an accelerating voltage of 30 kV.

## Results

### Preparation of WPCs

Wood fibers, HDPE, silane coupling agent (A187) and lubricant (paraffin wax) were prepared according to the design proportions, which were mixed at high speed for 15 min in a high-speed mixer (SHR-10A, Tonghe Rubber & Plastic Machinery Co., Ltd., Zhangjiagang, China). Table 3 details the specific proportioning scheme. Afterwards, the mixed materials were

**Table 3. Specimen proportioning scheme.**

| Specimen | Wood fiber (%) | HDPE (%) | A187 (%) | Paraffin wax (%) |
|---|---|---|---|---|
| *Specimen* | 50 | 45 | 4 | 1 |

pelletized and extrusion molded in a two-stage plastic extruder (SJSH30/SJ45, Nanjing Rubber & Plastic Machinery Plant, Nanjing, China), and then prepared into standard specimens with dimensions of 80 mm×13 mm×4 mm (L×W×T) for the bending test specimens and of 80 mm×10 mm×4 mm for the impact test specimens.

Temperature settings of single-screw extruder were as follows: 145˚C at zone 1, 155˚C at zone 2, 165˚C at zone 3, 165˚C at zone 4, and 165˚C at head area.

Temperature settings of double-screw extruder were as follows: 145˚C at zone 1, 150˚C at zone 2, 155˚C at zone 3, 150˚C at zone 4, 165˚C at zone 5, 165˚C at zone 6, 165˚C at zone 7, and 165˚C at zone 8.

Extrusion molding conditions: extruder barrel temperature 145–170˚C, head temperature 165˚C, main frequency 22 Hz, feeding speed 45 r/min, and pelletizing speed 356 r/min. Subsequently, the prepared WPCs were prepared into standard specimens using a small precision sliding table saw (HW110L-50, Hongrui Precision Cutting Tools Co., Ltd., Dongguan, China) and a dumbbell sampler (BGD-5102, Begeda Inspection Instruments Co., Ltd., Qingdao, China).

## Results of A187-modified wood fibers

After the impulse-cyclone drying + A187 treatment, results of A187-modified wood fibers as shown in Fig 2.

## Mechanical properties of UV-accelerated aged composite after wood fiber impulse-cyclone drying

**Flexural strength.**    Table 4 presents the changes in mechanical properties of HDPE/wood fiber composites after UV-accelerated aging (S2 Table in S1 File). The wood fibers in specimens 1–5 were modified by impulse-cyclone drying and A187 separately under the following conditions: 180˚C, 180˚C+A187, 200˚C+A187, 220˚C+A187, and 240˚C+A187.

The variation trends of the flexural strength of HDPE/wood fiber composites as shown in Fig 3.

**Flexural modulus.**    Table 5 illustrates the variation patterns of flexural moduli for HDPE/wood fiber composites (S3 Table in S1 File). The wood fibers in specimens 1–5 were treated by impulse-cyclone drying and modified with A187 separately under the following conditions: 180˚C, 180˚C + A187, 200˚C + A187, 220˚C + A187, and 240˚C + A187.

As indicated by the variation trends of flexural modulus in Fig 4.

**Impact strength.**    Table 6 lists the impact properties of HDPE/wood fiber composites before and after UV aging (S4 Table in S1 File). The wood fibers in specimens 1–5 were treated by impulse-cyclone drying and modified with A187 separately under the following conditions: 180˚C, 180˚C + A187, 200˚C + A187, 220˚C + A187, and 240˚C + A187. During the initial aging stage, i.e. within an aging period of 0–500 h, the impact performance of specimen 4 was slightly improved. However, the loss in impact performance began to increase as the aging time was extended.

According to the analysis of impact property variations in Fig 5.

## The surface chromaticity of UV-accelerated aged composites

Fig 6 displays the changes in surface chromaticity of HDPE/wood fiber composites with the time of UV-accelerated aging (S5 Table in S1 File). The wood fibers in specimens 1–5 were

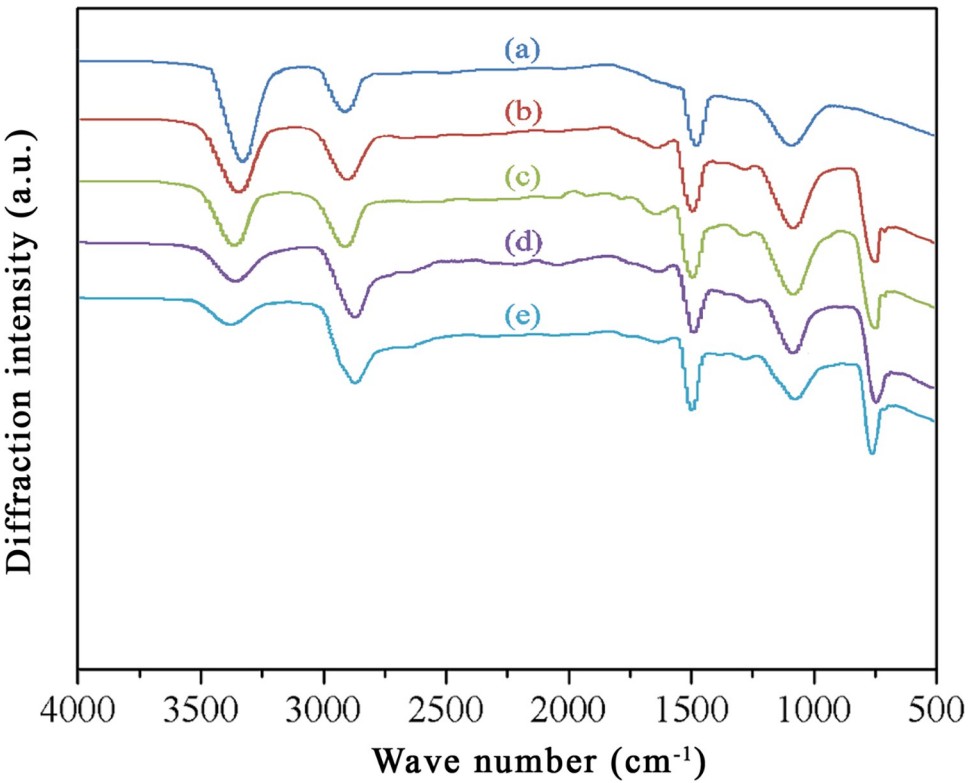

**Fig 2. FTIR spectra of wood fibers treated by high-temperature impulse-cyclone drying and modified by A187.** (a) 180˚C; (b) 180˚C+A187; (c) 200˚C+A187; (d) 220˚C+A187; (e) 240˚C+A187.

treated by impulse-cyclone drying and modified with A187 separately under the following conditions: 180˚C, 180˚C + A187, 200˚C + A187, 220˚C + A187, and 240˚C + A187.

## The surface microstructure of UV-accelerated aged composites

Fig 7 illustrates the surface microstructures of specimens after UV-accelerated aging (S2 Fig in S1 File). The wood fibers in specimens 1–5 were treated by impulse-cyclone drying and modified with A187 separately under the following conditions: 180˚C, 180˚C + A187, 200˚C + A187, 220˚C + A187, and 240˚C + A187.

## Discussion

### FTIR analysis of A187-modified wood fibers

Compared to the untreated wood fibers, the silanol produced by hydrolysis of coupling agent formed a covalent bond Si-O-C with the hydroxyl group of modified wood fibers, indicating

**Table 4. Flexural strengths before and after UV-accelerated aging.**

| Specimen No. | Flexural strength (MPa) | | | | | | |
|---|---|---|---|---|---|---|---|
| | **0h** | **500h** | **1000h** | **1500h** | **2000h** | **2500h** | **3000h** |
| *Specimen 1* | 53.16 | 52. 34 | 51. 96 | 50.46 | 47.37 | 44.49 | 43.15 |
| *Specimen 2* | 74.27 | 73.35 | 72.04 | 71.74 | 68.12 | 65.67 | 62.08 |
| *Specimen 3* | 75.82 | 74.68 | 73.71 | 72.73 | 69.42 | 66.38 | 63.45 |
| *Specimen 4* | 77.46 | 77.64 | 76.02 | 75.44 | 72.74 | 68.03 | 65.40 |
| *Specimen 5* | 71.14 | 70.04 | 69.46 | 68.36 | 65.06 | 62.84 | 59.63 |

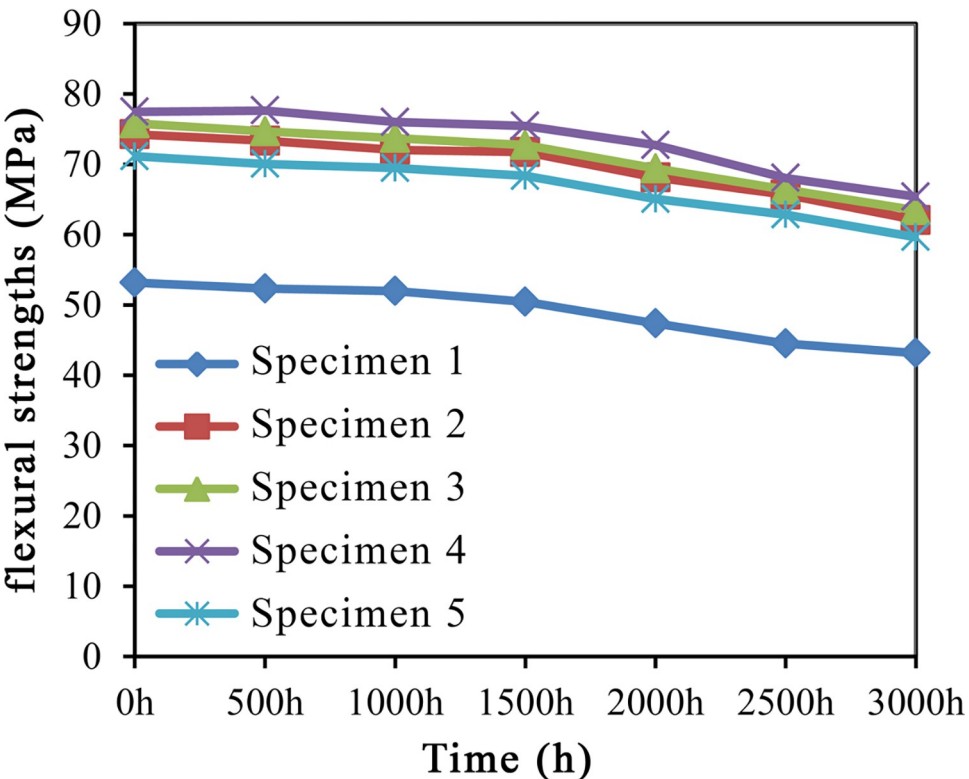

**Fig 3. Variation trends of flexural strengths.**

occurrence of polycondensation and grafting reactions between the A187 and the wood fibers. In the FTIR spectrum of A187-modified wood fibers, a new characteristic absorption peak appeared at 770 cm$^{-1}$, which was ascribed to the Si-O-Si bond produced by self-polymerization of silanol. Additionally, after the impulse-cyclone drying + A187 treatment, the hydroxyl absorption peak tended to weaken and decrease with rising airflow temperature. As shown in Fig 2, at the same drying temperatures, the alkoxy group in A187 reacted with the hydroxyl group of wood fibers, thereby effectively reducing the number of hydroxyl groups.

### Effects of wood fiber impulse-cyclone drying on the mechanical properties of UV-accelerated aged composites

**Flexural strength.** As is clear from Table 4, the mechanical properties of specimens modified by a combination of impulse-cyclone drying and A187 were superior to other matches. The comparison results of bending behavior relationship were: impulse-cyclone drying at

**Table 5. Flexural moduli before and after UV-accelerated aging.**

| Specimen No. | Flexural modulus (MPa) | | | | | | |
|---|---|---|---|---|---|---|---|
| | 0h | 500h | 1000h | 1500h | 2000h | 2500h | 3000h |
| Specimen 1 | 2427.01 | 2378.63 | 2299.66 | 2219.79 | 2087.37 | 1954.49 | 1822.15 |
| Specimen 2 | 2489.34 | 2483.45 | 2385.34 | 2286.99 | 2160.12 | 2033.67 | 1947.08 |
| Specimen 3 | 2525.37 | 2535.98 | 2442.92 | 2334.22 | 2217.61 | 2101.75 | 2004.17 |
| Specimen 4 | 2543.11 | 2548.38 | 2444.32 | 2334.43 | 2290.89 | 2146.20 | 2082.08 |
| Specimen 5 | 2469.84 | 2402.32 | 2358.93 | 2274.33 | 2121.16 | 2018.97 | 1914.20 |

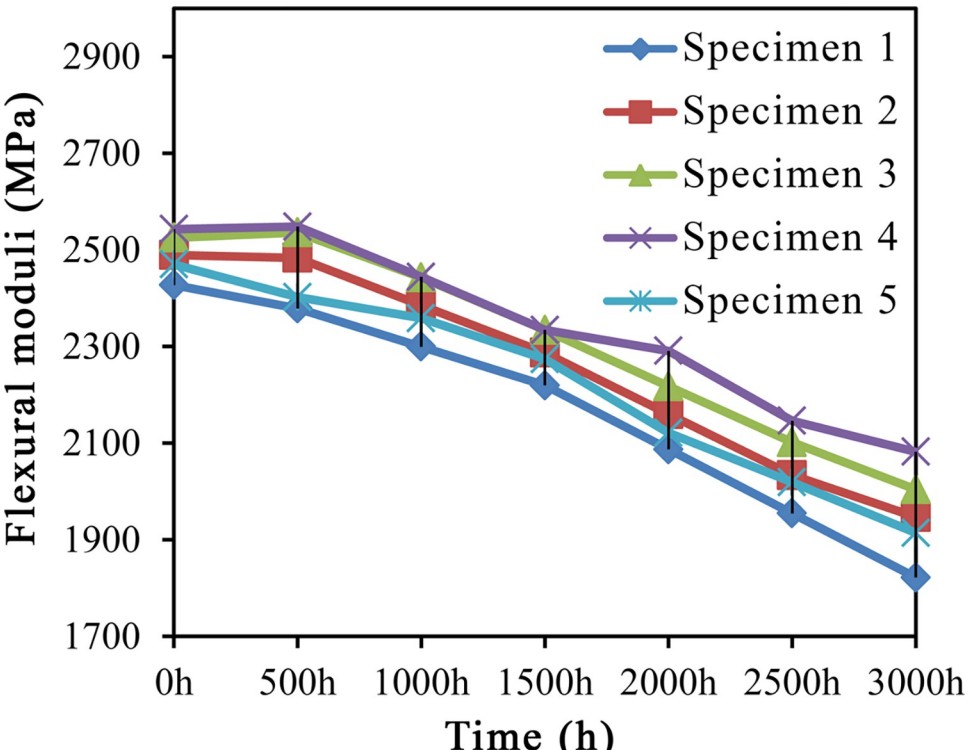

**Fig 4. Variation trends of flexural moduli.**

220˚C + A187> impulse-cyclone drying at 200˚C + MAPE> impulse-cyclone drying at 180˚C + A187> impulse-cyclone drying at 240˚C+ A187> impulse-cyclone drying at 180˚C.

Throughout the 0–3000 h aging process, the flexural strength of specimen 1 exhibited a continuous downward trend. In contrast, the flexural strengths of specimens 2–5 changed insignificantly over 0–1000 h, which began to decrease after 1500 h of UV-accelerated aging. At 1500 h following UV-accelerated aging, the flexural strength of specimen 4 dropped from 77.46 MPa to 75.44 MPa, which though remained as the highest value among the five groups of specimens. Meanwhile, the flexural strengths of specimens 2–5 were still higher than the specimen 1 that was aged for 0 h without A187 addition. The flexural strength of specimen 4 dropped to 65.40 MPa at 3000 h of UV-accelerated aging, despite still being the highest value among the five groups of specimens. As for specimens 2–5, their flexural strengths decreased by 16.41%, 16.32% 15.57%, and 16.17%, respectively. The specimen 1 without A187 addition exhibited the sharpest decrease (18.83%) in flexural strength.

**Table 6. Impact strengths before and after UV-accelerated aging.**

| Specimen No. | Impact strength (KJ·m$^{-2}$) | | | | | | |
|---|---|---|---|---|---|---|---|
| | **0h** | **500h** | **1000h** | **1500h** | **2000h** | **2500h** | **3000h** |
| *Specimen 1* | 11.29 | 11.04 | 10.97 | 10.73 | 10.47 | 10.21 | 9.95 |
| *Specimen 2* | 13.04 | 13.12 | 13.02 | 12.50 | 12.05 | 11.20 | 11.65 |
| *Specimen 3* | 13.91 | 13.47 | 13.12 | 13.33 | 13.04 | 12.77 | 12.47 |
| *Specimen 4* | 14.30 | 14.40 | 14.32 | 13.72 | 13.44 | 13.13 | 12.85 |
| *Specimen 5* | 11.61 | 11.48 | 11.28 | 11.09 | 10.87 | 10.59 | 10.25 |

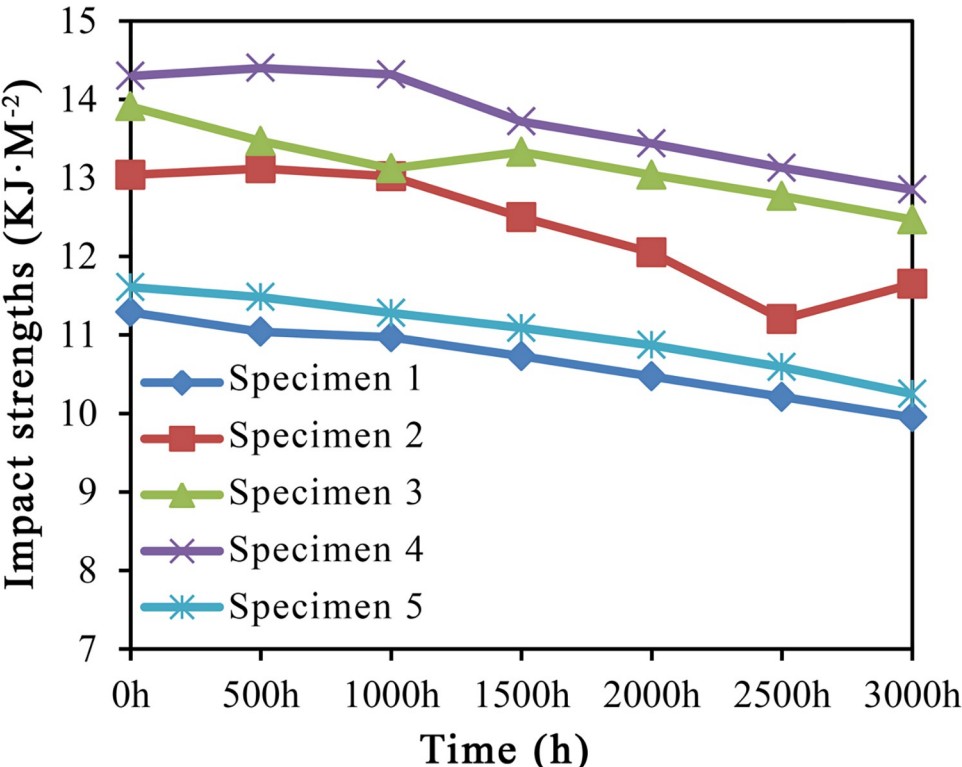

**Fig 5. Variation trends of impact strengths.**

The foregoing results indicated that the impulse-cyclone drying of wood fibers led to a decrease in hydroxyl groups. The A187 underwent an esterification reaction with the free hydroxyl groups on the wood fibers. Meanwhile, the non-polar macromolecular chains on the other end were physically entangled and connected with the HDPE matrix, which acted as a bridge to improve the compatibility between HDPE and wood fibers, thereby enhancing the mechanical properties of WPCs.

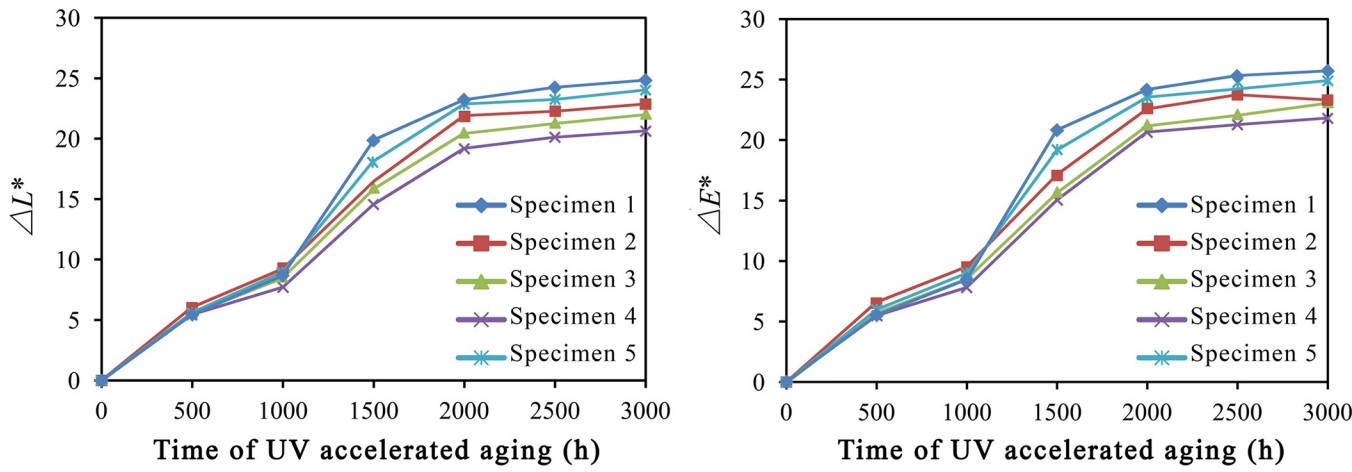

**Fig 6. Surface chromaticity changes after UV-accelerated aging.** $\Delta L^*$ - Luminosity value; $\Delta E^*$ - Degree of discoloration.

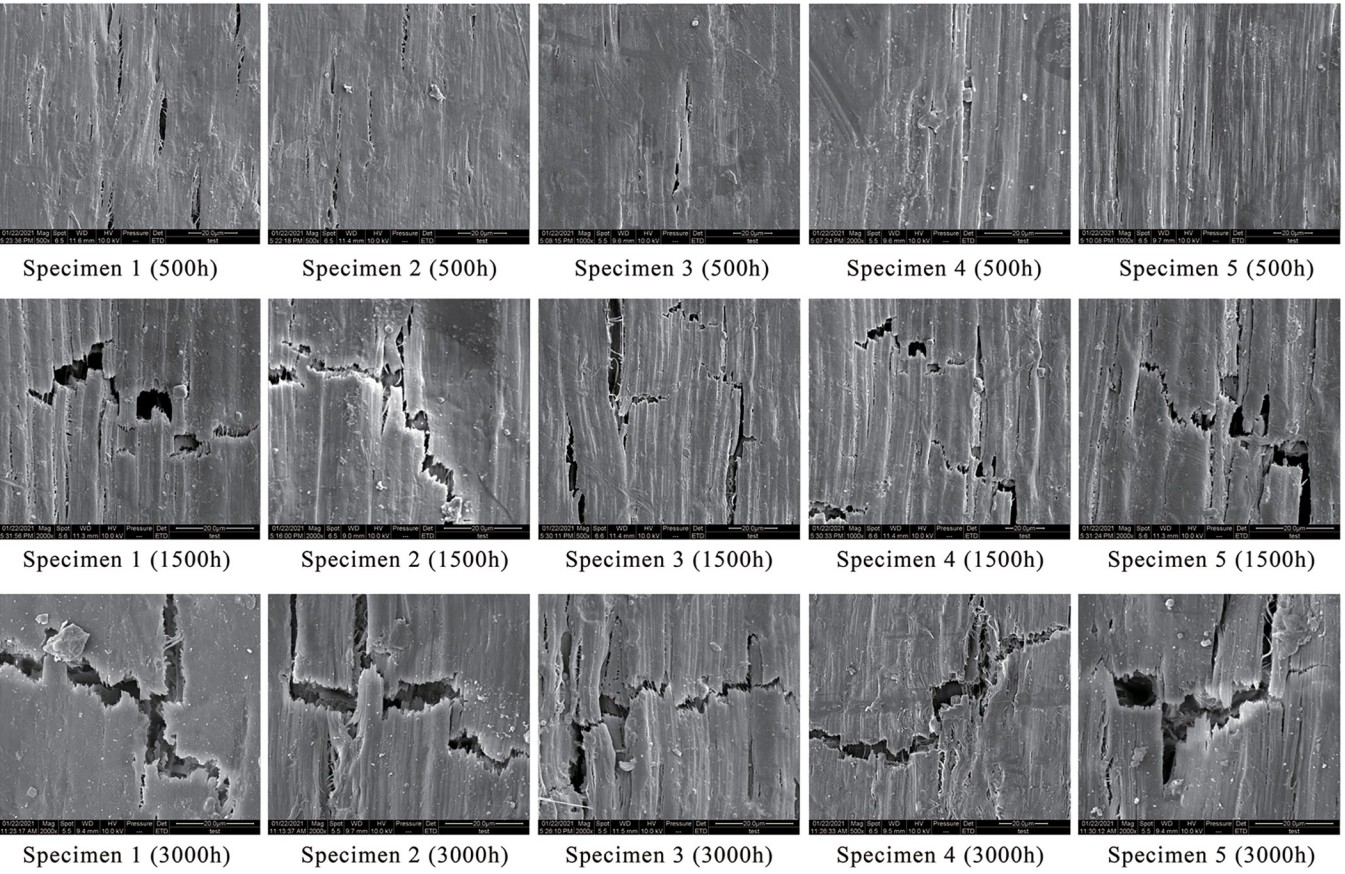

Specimen 1 (500h) Specimen 2 (500h) Specimen 3 (500h) Specimen 4 (500h) Specimen 5 (500h)

Specimen 1 (1500h) Specimen 2 (1500h) Specimen 3 (1500h) Specimen 4 (1500h) Specimen 5 (1500h)

Specimen 1 (3000h) Specimen 2 (3000h) Specimen 3 (3000h) Specimen 4 (3000h) Specimen 5 (3000h)

**Fig 7. Surface microstructures after UV-accelerated aging(Mag-2000X).**

Besides, the impulse-cyclone drying treatment also led to reduced polarity of wood fibers and improved crystallinity of cellulose. When treated at 220˚C, the fiber polarity decreased most evidently, so that the wood-plastic bonding interface was most compatible, and the mechanical properties were the best. After 3000 h of UV aging, the flexural strength was 65.40 MPa. At impulse-cyclone drying temperatures of 180˚C and 220˚C, enhanced flexural strength of HDPE/wood fiber composites was noted between 0–500 h. This was because by the action of temperature, the A187 promoted the enhancement of rigid structure in the HDPE/wood fiber composites, causing a temporary improvement in the aging resistance, which then began to degrade.

According to the variation trends in Fig 3, the flexural strength of HDPE/wood fiber composites tended to decrease with the prolongation of aging time. This suggested that treatment of wood fibers by impulse-cyclone drying and A187 modification exerted a protective effect on the HDPE/wood fiber composites. The evenly distributed wood fibers on the composite surface prevented the direct contact and absorption of UV light by HDPE to a certain extent. Specimen 4 exhibited better effect than other specimens, which was more capable of protecting the flexural strength of HDPE/wood fiber composites. Optimal flexural strength of HDPE/wood fiber composites was achieved with the wood fibers treated by 220˚C impulse-cyclone drying + A187.

**Flexural modulus.** UV-accelerated aging caused decreases in the flexural moduli of all specimens to varying degrees. At 3000 h after UV aging, the flexural modulus of specimen 4

was evidently higher than that of other specimens. After addition of A187, specimens 3 and 4 exhibited slight improvements in the flexural modulus within an aging period of 0–500 h. However, the losses in flexural modulus began to increase with the prolongation of aging time, and within 1500–3000 h after UV aging, the flexural moduli showed an overall trend of continuous decrease.

With the prolongation of aging time, the flexural moduli of HDPE/wood fiber composites exhibited a downward trend in Fig 4. Prior to UV aging, the flexural modulus of specimen 1 was 2427.01 MPa, which was the lowest value among the five groups of specimens. After aging for 500 h, the loss in flexural modulus of specimen 1 was 1.9%. As the aging time increased, the losses in flexural modulus began to increase. The flexural modulus of specimen 4 decreased by 3.88% at 1000 h following UV-accelerated aging, which was still higher than that of specimen 1 aged for 0 h. After 3000 h of UV aging, the flexural moduli of specimens 2–5 decreased by 21.78%, 20.64%, 18.13% and 22.49%, respectively, while the specimen 1 exhibited the sharpest decrease (24.92%) in flexural modulus.

It was concluded that the addition of A187 increased the flexural modulus of HDPE/wood fiber composites. UV light caused certain damage to the composite surfaces, leading to appearance of surface cracks, so that the flexural moduli of all specimens showed an overall decreasing trend. The reason was that the damage to the surface layer caused by UV-accelerated aging somewhat weakened the flexural properties of HDPE/wood fiber composites. Apart from reducing the flexural modulus, the cracks on composite surfaces also reduced the efficiency of stress transmission inside the composite material, thereby degrading the mechanical properties of the composites.

**Impact strength.**   The impact performance of HDPE/wood fiber composites was effectively enhanced after impulse-cyclone drying and A187 modification of wood fibers. At 0 h of aging, the impact strength of specimen 1 was 11.29 KJ·m$^{-2}$, which was the lowest value among the five specimens. After 500 h of aging, the impact strength of specimen 4 increased by 0.6%, while the flexural moduli of other specimens showed an overall downward trend.

At 1000 h following UV-accelerated aging, the impact strength of specimen 4 was 14.32 KJ·m$^{-2}$. Since the free radical short-chain crosslinking in the thermoplastic matrix of WPCs was a short-term phenomenon, the surface cracks of HDPE/wood fiber composites gradually increased with continuation of time despite initial slight improvement in the impact strength, eventually leading to degraded impact performance.

After 3000 h of UV-accelerated aging, the flexural strengths of specimens 2–5 decreased by 10.66%, 10.35%, 10.13% and 11.71%, while specimen 1 showed an 11.87% reduction. At this time, the impact strengths of specimens 3 and 4 were 12.47 KJ·m$^{-2}$ and 12.85 KJ·m$^{-2}$, which were still higher than the specimen 1 aged for 0 h. Thus, clearly, the addition of A187 effectively enhanced the impact strength of UV-aged HDPE/wood fiber composites. Owing to the wood fibers modified by 220˚C impulse-cyclone drying + A187, an optimal state of composite impact strength could be attained.

## Effects of modified wood fibers on the surface chromaticity of UV-accelerated aged composites

As indicated by the Fig 6, after adding wood fibers modified under the foregoing conditions, the variation trends of luminosity and chromatic difference were almost identical among the HDPE/wood fiber composites. The $\Delta L^*$ and $\Delta E^*$ values increased over the aging time, and the surface discoloration of five specimens was rather evident.

Within 0–1000 h after UV aging, the changes in $\Delta L^*$ and $\Delta E^*$ values were rather small. Larger increases in $\Delta L^*$ and $\Delta E^*$ values were observed within 1000–2000 h after UV aging.

After 2000 h of UV aging, the $\Delta E^*$ value still increased, which though remained stable overall. After 3000 h of UV aging, the surface chromaticity of specimen 1 was grayer and whiter than the specimens 2–5. The specimen 4 exhibited $\Delta L^*$ and $\Delta E^*$ values of 20.636 and 21.845, which had the least change in surface chromatic difference.

Test results revealed an overall trend of rapid decline in chromophoric functional groups following a short-term activity, so that the surface discoloration of HDPE/wood fiber composites was primarily manifested as whitening. This was attributed to the chain scission of HDPE on the composite surface, which led to decreased composite density at the entangled part of amorphous region through photodegradation reaction, consequently increasing the surface luminosity of the composites. However, after addition of wood fibers treated by impulse-cyclone drying and A187 modification, the photodegradation reaction was effectively prevented, suggesting the role of modified wood fibers in resisting the UV aging and protecting the surface chromaticity.

## Effects of modified wood fibers on the surface microstructure of UV-accelerated aged composites

At 500 h following UV aging, no obvious cracks were found on the surfaces of five specimens. Specimens 3 and 4 exhibited the flattest and smoothest surfaces. After 1500 h of UV aging, the surface cracks of HDPE/wood fiber composites enlarged. Specimens 3 and 4 exhibited markedly less cracks than other specimens, while specimen 1 had the largest cracks. After 3000 h of UV aging, the surface cracks of specimens further increased as the UV aging prolonged, which were numerous in number. Comparatively, the surface cracks of specimen 4 were smaller, revealing the best aging resistance.

The results demonstrated that after impulse-cyclone drying at 220˚C, the wood fibers were evenly dispersed in the interior and surface of composites. After A187 treatment, one end of the A187 functional groups reacted with the hydroxyl groups, while the other end chemically reacted or physically entangled with the functional groups in HDPE, thereby linking the wood fibers and HDPE effectively to block certain amounts of UV light. In this way, the UV light was incapable of tearing the HDPE/wood fiber composites, which was converted into heat energy and dispersed. Meanwhile, this also decelerated the HDPE degradation and delayed the composite degradation, so that tiny cracks appeared on the specimen surfaces to improve the aging resistance of the composites.

More surface cracks were produced in the specimen 5 than the specimen 4, indicating that at excessively high temperature of impulse-cyclone drying, the hydroxyl groups of wood fibers were substantially reduced, so that there were insufficient hydroxyl groups to link the A187, leading to weakened bridging effect. As a result, the junction between wood fibers and HDPE was exposed, which could not block the UV light effectively, and degraded under the UV light, ultimately resulting in larger cracking.

## Conclusions

In this study, the impulse-cyclone drying and A187 modification of wood fibers are explored, and HDPE/wood fiber composites are fabricated using the modified wood fibers. The effects of wood fibers on the UV-accelerated aging characteristics of the composites are investigated. The research results are as follows:

FTIR analysis indicates that the A187 chemically reacted with the hydroxyl groups on the wood fiber surfaces. The silanol produced from hydrolysis of A187 and the covalent bond Si—O—C formed from hydroxyl groups of wood fibers affected the surface components of wood fibers to some extents. At the same drying temperatures, the alkoxy groups in A187 reacted

with the hydroxyl groups of wood fibers, thereby effectively reducing the number of hydroxyl groups.

UV-accelerated aging test reveals that the flexural strength, flexural modulus and impact strength of the composites tended to decrease continuously over aging time. After aging for 3000 h, specimen 4 exhibited the lease losses in mechanical properties, with flexural strength, flexural modulus and impact strength of 65.40 Mpa, 2082.08 Mpa, and 12.85 Mpa, respectively. During the 0–3000 h process of UV aging, the addition of wood fibers treated by impulse-cyclone drying and A187 modification led to an increase-then-decrease trend of specimens within a period of time. This suggests that the modified wood fibers can absorb part of the UV light to effectively reduce the surface oxidation caused by UV aging. The aging index of specimen 4 always remained higher than that of other specimens.

Spectrophotometer test demonstrates that the $\Delta L^*$ and $\Delta E^*$ values increased with the prolongation of UV aging time. The $\Delta E^*$ value was fundamentally dependent on the $\Delta EEL^*$ value. At the 0–1000 h stage of aging, the increases in $\Delta L^*$ and $\Delta E^*$ values were rather drastic, which then tended to stabilize after 1000 h. Specimen 4 exhibited smaller changes in the degree of discoloration.

According to the surface microstructure observation following UV-accelerated aging, no obvious cracks were present on the surfaces of five specimens after 500 h of UV aging. At 1500 h following UV aging, the surface cracks of HDPE/wood fiber composites enlarged. Specimens 3 and 4 exhibited markedly less cracks than other specimens, while specimen 1 had the largest surface cracks. After 3000 h of UV aging, the cracks on specimen surfaces further increased as the UV aging continued, which were numerous in number. Comparatively, the surface cracks of specimen 4 were smaller, showing the best aging resistance.

In conclusion, the addition of wood fibers modified by 220˚C impulse-cyclone drying and A187 can enhance the UV aging resistance and thermal stability of WPCs, which delays the surface oxidation of HDPE/wood fiber composites and improves the aging resistance of WPCs in terms of surface discoloration and mechanical properties. However, such enhancing effects turn to decline when the temperature of impulse-cyclone drying is excessively high.

## Supporting information

**S1 File.**
(DOCX)

## Author Contributions

**Writing – original draft:** Qingde Li.

**Writing – review & editing:** Feng Chen, Tonghui Sang.

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
