## [Decision Letter · Decision Letter 0]

10 Dec 2021

PONE-D-21-36767Effects of Wood Fiber Impulse-Cyclone Drying Process on the UV-Accelerated Aging Properties of Wood-Plastic CompositesPLOS ONE

Dear Dr. Chen,

Thank you for submitting your manuscript to PLOS ONE. After careful consideration, we feel that it has merit but does not fully meet PLOS ONE’s publication criteria as it currently stands. Therefore, we invite you to submit a revised version of the manuscript that addresses the points raised during the review process.

The paper is deserve publication in PLOS ONE given the positive response of the respected reviewers after minor revision.

We look forward to receiving your revised manuscript.

Kind regards,

Khalil Abdelrazek Khalil, Ph.D.

Academic Editor

PLOS ONE

Journal Requirements:

"The National Natural Science Foundation of China (Grant No.31901243) financially supported this

research. Qingde Li conceived and designed the experiments; Feng Chen performed the experiments;

Tonghui Sang analyzed the data and wrote the paper."

Additional Editor Comments:

Pleas address the comments of the second reviewer and submit a revised version.

Reviewers' comments:

Reviewer's Responses to Questions

**Comments to the Author**

1. Is the manuscript technically sound, and do the data support the conclusions?

Reviewer #1: Yes

Reviewer #2: Partly

2. Has the statistical analysis been performed appropriately and rigorously? 

Reviewer #1: Yes

Reviewer #2: Yes

3. Have the authors made all data underlying the findings in their manuscript fully available?

Reviewer #1: Yes

Reviewer #2: Yes

4. Is the manuscript presented in an intelligible fashion and written in standard English?

Reviewer #1: Yes

Reviewer #2: Yes

5. Review Comments to the Author

Reviewer #1: This study focused on WPC manufactured by pulse cyclone dryer, studied the effect of high temperature hot air treatment conditions on the modification efficiency of biomass fiber, and discussed the effects of fiber modification conditions on UV accelerated aging, mechanical properties, chromaticity, dynamic thermomechanical (DMA) properties and thermogravimetric (TG) behavior of WPC. The results showed that the UV aging resistance and thermal stability of WPC could be improved by adding wood fiber modified by 220 ° C pulse cyclone drying and A187. However, when the pulse cyclone drying temperature is too high, the strengthening effect will decrease. In general, the research method and process are relatively clear, the research data is full and accurate, and the research results answer the research questions.

Reviewer #2: 1. What is the purpose of this research? To improve the composites performance of biomass composites or the effect of high-temperature hot air treatment conditions on the efficiency of biomass fiber modification, or something else?

2. Please discuss the scientific nature of the experimental method.

3. The literature review does not fully explain the advancement and innovation of this research.

4.The abstract cannot be replaced by a conclusion.

6. PLOS authors have the option to publish the peer review history of their article (what does this mean?). If published, this will include your full peer review and any attached files.

Reviewer #1: No

Reviewer #2: No

---

## [Author Response · Author response to Decision Letter 0]

15 Feb 2022

Response to reviewers’ comments

Dear Managing Editor,

Thanks so much for your professional review of our manuscript entitled “ Effects of Wood Fiber Impulse-Cyclone Drying Process on the UV-Accelerated Aging Properties of Wood-Plastic Composites ” (ID:PONE-D-21-36767). We also highly appreciate the beneficial suggestions and comments from the reviewers. All the questions pointed out by reviewers have been answered carefully and discussed in detail. The changes are marked with GREEN font in the revised manuscript. The main corrections in the paper and the responses to the reviewer’s comments are listed as follows in the “Responses to the Reviewers”. We hope that these revisions are satisfactory and that the revised version is now suitable for publication in PLOS ONE. 

Thank you very much for your work concerning our article. 

Sincerely yours,

Qingde Li

Journal Requirements:

Response: Thank you for this good suggestion. We have eliminated it and the changes can be found in the revised manuscript.

2. Thank you for stating the following in the Acknowledgments Section of your manuscript: "The National Natural Science Foundation of China (Grant No.31901243) financially supported this research. Qingde Li conceived and designed the experiments; Feng Chen performed the experiments; Tonghui Sang analyzed the data and wrote the paper."We note that you have provided funding information that is not currently declared in your Funding Statement. However, funding information should not appear in the Acknowledgments section or other areas of your manuscript. We will only publish funding information present in the Funding Statement section of the online submission form. Please remove any funding-related text from the manuscript and let us know how you would like to update your Funding Statement. Currently, your Funding Statement reads as follows: "The author(s) received no specific funding for this work." Please include your amended statements within your cover letter; we will change the online submission form on your behalf.

Response: Thank you for this good suggestion. We have eliminated it and the changes can be found on Page 13 in the revised manuscript.

Response: Thank you for this good suggestion. We have eliminated it and the changes can be found in the revised manuscript.

Review Comments to the Author：

Reviewer #1: This study focused on WPC manufactured by pulse cyclone dryer, studied the effect of high temperature hot air treatment conditions on the modification efficiency of biomass fiber, and discussed the effects of fiber modification conditions on UV accelerated aging, mechanical properties, chromaticity, dynamic thermomechanical (DMA) properties and thermogravimetric (TG) behavior of WPC. The results showed that the UV aging resistance and thermal stability of WPC could be improved by adding wood fiber modified by 220 ° C pulse cyclone drying and A187. However, when the pulse cyclone drying temperature is too high, the strengthening effect will decrease. In general, the research method and process are relatively clear, the research data is full and accurate, and the research results answer the research questions.

Response: Thank you for your congenial suggestion. We will refine our manuscript even better.

Reviewer #2: 

1. What is the purpose of this research? To improve the composites performance of biomass composites or the effect of high-temperature hot air treatment conditions on the efficiency of biomass fiber modification, or something else?

Response: Thank you for your congenial suggestion. It is the purpose of this research that the composites performance of biomass composites is improved through high-temperature hot air treatment.

2. Please discuss the scientific nature of the experimental method.

Response: Thank you for this good suggestion. A spectrophotometer (CM-2300d, Konica Minolta, Japan) was utilized to measure the surface chromaticity values of HDPE/wood fiber composite specimens. The L*a*b* color system developed by the International Commission of Illumination CIE (1976) was used for color notation. A universal mechanical tester (WDW-20, Kexin Testing Instrument, Changchun, Jilin, China) was utilized to measure the mechanical properties of HDPE/wood fiber composites before and after accelerated aging. The specimens test in accordance with ASTM D 790 and ASTM D256 . The test results were expressed as the arithmetic means of five specimens. A scanning electron microscope (QUANTA200, FEI, Netherlands) was utilized for the material surface characterization and the test results analysis. The working principle of scanning electron microscope (SEM) was based on the interaction between electrons and matter. The experimental process was implemented in accordance with ASTM E1588 (2017) , and the sample morphological characteristics were observed using QUANTA 200 SEM system under an accelerating voltage of 30 kV. The experimental methods in the article are scientifically carried out in accordance with international standards.

3. The literature review does not fully explain the advancement and innovation of this research.

Response: Thank you for pointing this out. The impulse-cyclone drying equipment was independently developed by us. This is where the innovation of the article. The article demonstrates the effect of impulse-cyclone drying technique through relevant experiments.

4.The abstract cannot be replaced by a conclusion.

Response: Thank you for your kind suggestion. we have revised it and the changes can be found in the abstract.

---

## [Decision Letter · Decision Letter 1]

28 Mar 2022

Effects of Wood Fiber Impulse-Cyclone Drying Process on the UV-Accelerated Aging Properties of Wood-Plastic Composites

PONE-D-21-36767R1

Dear Dr. Chen,

We’re pleased to inform you that your manuscript has been judged scientifically suitable for publication and will be formally accepted for publication once it meets all outstanding technical requirements.

Kind regards,

Khalil Abdelrazek Khalil, Ph.D.

Academic Editor

PLOS ONE

Additional Editor Comments (optional):

Thank you for addressing the reviewers comments

Reviewers' comments:

Reviewer's Responses to Questions

**Comments to the Author**

1. If the authors have adequately addressed your comments raised in a previous round of review and you feel that this manuscript is now acceptable for publication, you may indicate that here to bypass the “Comments to the Author” section, enter your conflict of interest statement in the “Confidential to Editor” section, and submit your "Accept" recommendation.

Reviewer #2: (No Response)

2. Is the manuscript technically sound, and do the data support the conclusions?

Reviewer #2: (No Response)

3. Has the statistical analysis been performed appropriately and rigorously? 

Reviewer #2: (No Response)

4. Have the authors made all data underlying the findings in their manuscript fully available?

Reviewer #2: (No Response)

5. Is the manuscript presented in an intelligible fashion and written in standard English?

Reviewer #2: (No Response)

6. Review Comments to the Author

Reviewer #2: (No Response)

7. PLOS authors have the option to publish the peer review history of their article (what does this mean?). If published, this will include your full peer review and any attached files.

Reviewer #2: No

---

## [Editor Report · Acceptance letter]

15 Sep 2022

PONE-D-21-36767R1 

Effects of Wood Fiber Impulse-Cyclone Drying Process on the UV-Accelerated Aging Properties of Wood-Plastic Composites 

Dear Dr. Chen:

I'm pleased to inform you that your manuscript has been deemed suitable for publication in PLOS ONE. Congratulations! Your manuscript is now with our production department. 

Kind regards, 

on behalf of

Dr. Khalil Abdelrazek Khalil 

Academic Editor

PLOS ONE